# Peer review of "Evaluation of performance of leading algorithms for variant pathogenicity predictions and designing a combinatory predictor method: application to Rett syndrome variants"

_PeerJ, doi:10.7717/peerj.8106_

## Round 0.1 · original submission · Major Revisions

Reviewers raised significant concern regarding methodologies of the paper, especially on the construction of the ensemble classifier. Please fully address these concerns. Reviewer 1 raised major concerns regarding the validity of findings, which is the crucial part for the publication of this paper.

Reviewer 1 ·

Basic reporting

a. The overall manuscript is not difficult to follow, however, some sentences and terms used are still confusing. For example, they used meta predictors and empirical in-silico predictors interchangeably.
b. Some mis-spelled words are present which is disturbing for the audience to read, such as in line 125 ‘tis’ and line 138 ‘photogenic’.
c. Some references are not in the same style as the rest, for example, line 220 ‘Li et al.[24]’. Also it was not correctly indexed. Some material lacks reference. For example, line 69, ‘ACMG recommendations’.

Experimental design

a. The design of the experiment is standard. The research lacks novelty and the idea of ensemble scores have long been adopted in the field.
b. The methods described are not clear enough for the audience to understand the logic and steps behind the actions. For example, 1) how to decide which predictor outperforms another based on sensitivity, specificity, accuracy and MCC altogether? 2) How are the different cut-off points for pathogenicity decided and why there are NAs in them?

Validity of the findings

One major concern regarding the quality of the manuscript is their method. The rationale to select top four predictors in each category to form their ensemble predictor is not clear. It is possible that other combinations of predictors would yield better result, but the authors presented no such information for the audience to judge their method. Also, ClinVar and HGMD (used as training data by many predictors being compared) have overlapping SNVs, and the study did not take this into consideration which would lead to bias in their conclusion. In sum, the conclusion is not well-supported by the study design and results.

Reviewer 2 ·

Basic reporting

see General comments for the author

Experimental design

see General comments for the author

Validity of the findings

no comment

Additional comments

This paper describes the selection of a suitable subset of the clinvar database and the use of a machine-learning method to combine the results from eight predictive programs to predict the clinical consequence of mutations. They started with twelve predictive programs and selected the best eight for their method. The method appeared to perform better than using each program alone or to using only two methods as in a previous study. They then used the method to predict the clinical outcome of the mutations that were previously labelled as possibly benign, possibly pathogenic, or variants with unknown consequences. The findings should interest scientists in the field.

The authors might want to address the following issues before publication:
- The best method used appear to be random forest but inconsistent statements occur earlier in the manuscript. For example, “Purpose: We implemented feature selection and support vector machine (SVM) classification methods”
- How were the results in Fig. 1 obtained? Did they run the twelve predictive programs themselves? Or the results came come https://sites.google.com/site/jpopgen/dbNSFP?
- Were the results in Fig 2 averages from the independent predictors and averages from the ensemble predictors?
- Why were the results for random forest from Table 5 and Table 6 different?
- In table 5: recall is the same as sensitivity: can remove one of them.
- There are a number of grammatical mistakes in the manuscript. The authors might want to proof-read the article.

Reviewer 3 ·

Basic reporting

No comment, the article is very well written and background and aims are both well established with sufficient references

Experimental design

No comment

Validity of the findings

No comment

Additional comments

The authors performed classifications of pathogenic and benign variants using multiple predictors by selecting the best combination. In general, the manuscript is well written and rational was very nicely explained. Since there are multiple variant scores both independent and ensemble, and they are all widely used to annotate variants, this manuscript would be a great resource to refer to when one would like to compare multiple scores and their performance. However I have some concerns in the decision of the feature selection and application of the model. My specific comments are below.

1. I see that the authors evaluated each predictor to rank them. But I did not get why top 4 from each independent and ensemble are the best combination. It could be top 3 or only top 4 from ensemble predictors may perform better than including independent predictors. If the authors tested other combinations, those results should be mentioned in the manuscript. Otherwise, it needs to be clarified how and why they decided to take top 4.

2. Authors used known pathogenic and benign variants from Rett syndrome to validate their model. Those are the only independent datasets used in the manuscript. Even though the authors showed the high performance of the model with the cross-validations, the number of variants used in the application for Rett syndrome was considerably small, especially there was only a single benign variant. It would greatly improve the validation if there were larger independent sets to test.

3. In practice, if someone wants to apply the model proposed in this manuscript, they would have to re-do what authors did (i.e., train the model again). I believe it should be able to share the pre-trained model from Weka pipeline. The authors should consider depositing the pre-trained model into a free online repository.

Minor comments
1. For fig. 1, it would be more readable if independent and ensemble predictors are labeled and also ordered by the rank (so that it’s easy to see which ones are used for the final model)
2. For fig. 2, the legend in the figure is “Meta” predictor while it is “ensemble” in the legend text. Not limited to the fig. 2 but across the manuscript, it is better to harmonize to one of the other.
3. Typo in line 125, “this reduced datasets tis” should be “this reduced dataset is”
4. Typo in line 135, “as like benign/pathogenic” should be “as likely benign/pathogenic”

---

## Round 0.2 · Major Revisions

The major point by reviewer 1 should be addressed before this manuscript can be accepted for publication.

Reviewer 1 ·

Basic reporting

No further comments.

Experimental design

I am not totally convinced by the authors' response but it's up to the editor to decide.

Validity of the findings

I don't think the authors addressed my question regarding the overlapping SNVs.

Reviewer 2 ·

Basic reporting

satisfactory.

Experimental design

satisfactory.

Validity of the findings

satisfactory

Additional comments

the manuscript has been improved.

Reviewer 3 ·

Basic reporting

No comment

Experimental design

No comment

Validity of the findings

No comment

Additional comments

My questions/concerns were sufficiently answered and I have no further comment

---

## Round 0.3 · accepted · Accept

Please note that PeerJ does not provide editorial services as standard so please make sure that everything on your manuscript is publication-ready.